# Traumatic Proximal Femoral Fractures during COVID-19 Pandemic in the US: An ACS NSQIP^®^ Analysis

**DOI:** 10.3390/jcm11226778

**Published:** 2022-11-16

**Authors:** Muhammad Umar Jawad, Connor M. Delman, Sean T. Campbell, Ellen P. Fitzpatrick, Gillian L. S. Soles, Mark A. Lee, R. Lor Randall, Steven W. Thorpe

**Affiliations:** 1Department of Orthopedic Surgery, Good Samaritan Regional Medical Center, Corvallis, OR 97330, USA; 2Department of Orthopaedic Surgery, University of California-Davis, Sacramento, CA 95817, USA

**Keywords:** ACS NSQIP, COVID-19, proximal femur fracture, US national data

## Abstract

In order to determine the impact of COVID-19 on the treatment and outcomes in patients with proximal femoral fracture’s (PFF), we analyzed a national US sample. This is a retrospective review of American College of Surgery’s (ACS) National Surgical Quality Improvement Program (NSQIP) for patients with proximal femoral fractures. A total of 26,830 and 26,300 patients sustaining PFF and undergoing surgical treatment were sampled during 2019 and 2020, respectively. On multivariable logistic regression, patients were less likely to have ‘presence of non-healing wound’ (*p* < 0.001), functional status ‘independent’ (*p* = 0.012), undergo surgical procedures of ‘hemiarthroplasty’(*p* = 0.002) and ‘ORIF IT, Peritroch, Subtroch with plates and screws’ (*p* < 0.001) and to be ‘alive at 30-days post-op’ (*p* = 0.001) in 2020 as compared to 2019. Patients were more likely to have a case status ‘emergent’, ‘loss of ≥10% body weight’, discharge destination of ‘home’ (*p* < 0.001 for each) or ‘leaving against medical advice’ (*p* = 0.026), postoperative ‘acute renal failure (ARF)’ (*p* = 0.011), ‘myocardial infarction (MI)’ (*p* = 0.006), ‘pulmonary embolism (PE)’ (*p* = 0.047), and ‘deep venous thrombosis (DVT)’ (*p* = 0.049) in 2020 as compared to 2019. Patients sustaining PFF and undergoing surgical treatment during pandemic year 2020 differed significantly in preoperative characteristics and 30-day postoperative complications when compared to patients from the previous year.

## 1. Introduction

The first case of COVID-19 was diagnosed on 20 January 2020 in Washington State in the U.S. [1]. Shortly thereafter, the World Health Organization (WHO) declared COVID-19 a pandemic on 11 March 2020 [1]. The effect of COVID-19 pandemic on the health care system in the U.S. has been unprecedented. The Centers of Medicare and Medicaid Services (CMS) announced delays for all non-essential medical, surgical, and dental procedures during COVID-19 outbreak on 18 March 2020 at the White House Task Force press briefing [2]. 

Surgical treatment of a hip fracture is typically thought of as an urgent procedure. Delay in hip fracture care has been linked to increased morbidity and mortality [3,4,5]. However, providing timely care to patients with hip fractures during COVID-19 was fraught with multiple challenges for the strained U.S. healthcare system. One of the challenges was the diversion of logistic and personnel resources resulting in potential delays in surgery. 

There have been multiple recent publications from around the world, attempting to explore the impact of COVID-19 pandemic on hip fracture care. These include the single center review of medical records [6,7,8,9,10], review of national claims databases [11,12], meta-analyses [13,14] and national database [15,16,17] consortium data reviews [18,19,20]. There have been multiple reports of higher early mortality and post op complications in hip fracture patients with COVID-19 infection [6,18,19,21,22].

Studies emanating from single centers suffer from inherent selection bias. Insurance claims data is limited, inconsistent and subject to interpretation [23]. Previous early reviews of national databases or consortium data lacked some of the key information such as potential delay in surgeries or a comparison group of patients from pre-COVID era [15,16,18,19].

The purpose of this study was to compare the pre-operative characteristics, 30-day postoperative complications and mortality for patients who were treated for a traumatic hip fracture in 2020 (during the COVID-19 pandemic) with those treated in 2019 (pre-COVID-19), using American College of Surgeon (ACS) National Surgical Quality Improvement Program (NSQIP). Secondarily, we sought to compare the time from hospital admission to surgical fixation between these groups.

## 2. Methods

This is a retrospective review of ACS NSQIP^®^ proximal femoral fracture dataset from 2019 and 2020. The inclusion criteria have been described below under the case selection. Patient with missing information were excluded from the respective analysis. The study was deemed exempt from institutional review board (IRB) review.

### 2.1. ACS-NSQIP^®^

The American College of Surgeons (ACS) National Surgical Quality Improvement Program (NSQIP)^®^ [23] was used to extract the data for patients undergoing surgical treatment of proximal femur fractures in 2020. Data for patients with the same ICD 10 diagnostic codes from 2019 were used for comparison. The NSQIP database is a collection of patient records from over 680 participating hospitals across the United States [23]. Each institution assigns a trained data reviewer who randomly selects from a variety of surgical cases and uploads deidentified patient information and outcomes up to 30 days post-op onto a Health Insurance Portability and Accountability Act (HIPAA) compliant web-based platform [23]. The data is made available to investigators affiliated with the participating hospitals. NSQIP extracts the data directly from patient’s medical chart, and not insurance claims. The data is risk-adjusted and case-mix adjusted [23]. The data from the entire year was used to counter balance any effect of seasonal variation in incidence, morbidity and mortality of hip fractures [24,25,26].

### 2.2. Case Selection

Case data were extracted from the yearly (2019 and 2020) Participant User File (PUF) provided by ACS. International Classification of Disease, 10th Revision (ICD-10) diagnosis codes were used to identify and extract cases with proximal femoral fractures: femoral neck (S72.0), intertrochanteric (S72.1) and subtrochanteric (S72.2). A detailed breakdown of all the cases with respective ICD-10 codes is provided in the Appendix A.

### 2.3. Pre-Operative and Intra-Operative Characteristics (Table 1)

Demographic characteristics including age, gender, race and ethnicity were extracted. Information regarding pre-existing medical conditions: diabetes (type 1, type 2 and non-diabetic), COPD (Chronic Obstructive Pulmonary Disease), ascites, CHF (Congestive Heart Failure), HTN (Hypertension), presence of non-healing wound, dyspnea, vent dependance, steroid use, loss of ≥10% of body weight, bleeding disorder, pre-operative transfusion, pneumonia, pre-operative UTI (Urinary tract infection) and pre-operative infection of surgical site (superficial, muscle fascia, bone/joint) were also extracted. In addition, NSQIP collects information regarding functional status of patients and smoking history. Functional status is categorized as ‘independent’, ‘partially dependent’ and ‘fully dependent’ based upon the ability of patient to perform activities of daily living (ADLs) such as clothing, bathing and toileting at their peak physical function during the 30 days prior to admission. Smoking history is defined as having smoked within a year prior to surgery.

**Table 1 jcm-11-06778-t001:** Cross Table Year of Diagnosis and Preoperative and Intraoperative Patient Characteristics with Traumatic Proximal Femoral Fractures.

Cross Table Year of Diagnosis and Preoperative and Intraoperative Patient Characteristics with Traumatic Proximal Femoral Fractures						
		2019		2020		
		n	Valid% of Total	n	Valid% of Total	*p*-Value
**Total Patients**		26,830	100	26,300	100	
Age						
	18–39 years	182	0.7	182	0.7	
	40–64 years	2661	9.9	2572	9.8	
	>65 years	23,987	89.4	23,546	89.5	0.853
Gender						
	Male	8684	32.4	8620	32.8	
	Female	18,145	67.6	17,677	67.2	
	Non-Bi	1	0	3	0	0.356
Race						
	White	17,426	91.2	16,003	89.8	
	African American	934	4.9	938	5.3	
	AIAN	108	0.6	99	0.6	
	API	641	3.4	766	4.3	
	Other	0	0	10	0.1	**<0.001**
Ethnicity						
	Non-Hispanic	17,802	93.2	16,973	93.4	
	Hispanic	1295	6.8	1206	6.6	0.57
Diabetes						
	Type 1	2155	8	2109	8	
	Type 2	2909	10.8	2916	11.1	
	Non-Diabetic	21,766	81.1	21,275	80.9	0.664
Smoking Status						
	Non-Smoker	23,418	87.3	22,937	87.2	
	Smoker	3412	12.7	3363	12.8	0.809
Dyspnea						
	At rest	241	0.9	221	0.8	
	At moderate exertion	2036	7.6	1904	7.2	
	No Dyspnea	24,553	91.5	24,175	91.9	0.231
Functional Status						
	Independent	20,885	78.5	20,118	77.2	
	Partially Dependent	4822	18.1	4986	19.1	
	Fully Dependent	888	3.3	967	3.7	**<0.001**
Ventilator Dependant						
	Yes	44	0.2	53	0.2	
	No	26,786	99.8	26,247	99.8	0.311
COPD						
	Yes	2802	10.4	2652	10.1	
	No	24,028	89.6	23,648	89.9	0.172
Ascites						
	Yes	71	0.3	73	0.3	
	No	26,759	99.7	26,227	99.7	0.774
CHF						
	Yes	1040	3.9	944	3.6	
	No	25,790	96.1	25,356	96.4	0.081
HTN						
	Yes	17,625	64.3	17,027	64.7	
	No	9565	35.7	9273	35.3	0.345
Disseminated Cancer						
	Yes	488	1.8	471	1.8	
	No	26,342	98.2	25,829	98.2	0.809
Presence of Wound						
	Yes	800	3	670	2.5	
	No	26,030	97	25,630	97.5	**0.002**
Steroid Use						
	Yes	1382	5.2	1287	4.9	
	No	25,448	94.8	25,013	95.1	0.174
≥10% loss of body weight						
	Yes	647	2.4	1054	4	
	No	26,183	97.6	25,246	96	**<0.001**
Bleeding Disorder						
	Yes	3935	14.7	3814	14.5	
	No	22,895	85.3	22,486	85.5	0.591
Pre-Operative Transfusion						
	Yes	895	3.3	951	3.6	
	No	25,935	96.7	25,349	96.4	0.078
Pre-Operative Sepsis						
	None	23,413	87.3	22,650	86.1	
	SIRS	3290	12.3	3511	13.3	
	Sepsis	116	0.4	126	0.5	
	Septic Shock	11	0	13	0	**0.002**
Emergent						
	Yes	8437	31.4	9017	34.3	
	No	18,393	68.6	17,283	65.7	**<0.001**
Pre-Operative Superficial Surgical Site Infection						
	Yes	5	0	2	0	
	No	26,825	100	26,298	100	0.268
Pre-Operative Deep Surgical Site Infection (Muscles/Fascia)						
	Yes	1	0	1	0	
	No	26,829	100	26,299	100	0.989
Pre-Operative Deep Organ Surgical Site Infection (Bone/ Joint)						
	Yes	3	0	5	0	
	No	26,827	100	26,295	100	0.462
Pneumonia						
	Yes	204	0.8	215	0.8	
	No	26,626	99.2	26,085	99.2	0.457
Pre-Operative Urinary Tract Infection						
	Yes	236	0.9	234	0.9	
	No	26,594	99.1	26,066	99.1	0.803
Wound Class						
	Clean	26,454	98.6	25,931	98.6	
	Clean/Contaminated	318	1.2	320	1.2	
	Contaminated	52	0.2	40	0.2	
	Dirty/Infected	6	0	9	0	0.551
ASA						
	Class I	253	0.9	229	0.9	
	Class II	4623	17.3	4280	16.3	
	Class III	16,503	61.6	16,247	61.9	
	Class IV	5348	20	5458	20.8	
	Class V	48	0.2	47	0.2	**0.014**
Anesthesia						
	General	19,095	71.2	17,867	67.9	
	Regional	5359	20	5800	22.1	
	Other	2373	8.8	2630	10	**<0.001**
Surgical Procedure						
	Hemiarthroplasty	2851	10.6	2527	9.6	
	Total Hip Arthroplasty	1561	5.8	1497	5.7	
	ORIF Neck or Prosthetic	7851	29.3	7855	29.9	
	ORIF IT, Peritroch, Subtroch with plates and screws	1862	6.9	1586	6	
	IM Nail for IT, Peritroch, Subtroch	11,772	43.9	11,883	45.2	
	Other	933	3.5	952	3.6	**<0.001**

The intraoperative characteristics analyzed included anesthesia (general, regional and other) and wound class (clean, clean/contaminated, contaminated, and dirty/infected). Information regarding ASA (American Society of Anesthesiologist) class was also extracted and presented in Table 1. NSQIP presents surgical procedures as CPT (Current Procedural Terminology) codes. Surgical procedures with frequency more than 1% have also been presented separately in Table 1. The category ‘others’ includes CPT codes with frequency less than 1%. A detailed account of all the CPT codes for the entire cohort is presented in Appendix A.

### 2.4. Thirty-Day Postoperative Characteristics and Mortality (Table 2)

A variety of 30-day postoperative characteristics are collected by NSQIP. These include acute renal failure, postoperative surgical site infection (superficial, muscle/fascia, bone/joint), wound dehiscence, pneumonia, reintubation, pulmonary embolism, ventilator >48 h, UTI, CVA, cardiac arrest, myocardial infarction, transfusion >2, DVT, sepsis, septic shock, return to operating room, readmission due to hip fracture and discharge destination. In addition, 30-day post-op mortality data is also presented in Table 2. In addition, continuous variables: length of total hospital stay, total operation time and time from admission to surgical fixation, have also been extracted and presented as mean and standard deviation (SD).

**Table 2 jcm-11-06778-t002:** Cross Table Year of Diagnosis and Postoperative Patient Characteristics with Traumatic Proximal Femoral Fractures.

Cross Table Year of Diagnosis and Postoperative Patient Characteristics with Traumatic Proximal Femoral Fractures						
		2019		2020		
		n	Valid% of Total	n	Valid% of Total	*p*-Value
**Total Patients**		26,830	100	26,300	100	
Acute Renal Failure						
	Yes	188	0.7	240	0.9	
	No	26,642	99.3	26,060	99.1	**0.006**
Postoperative Superficial Surgical Site Infection						
	Yes	271	1	296	1.1	
	No	26,559	99	26,004	98.9	0.196
Postoperative Deep Surgical Site Infection (Muscles/Fascia)						
	Yes	35	0.1	41	0.2	
	No	26,795	99.9	26,529	99.8	0.438
Postoperative Deep Organ Surgical Site Infection (Bone/Joint)						
	Yes	68	0.3	80	0.3	
	No	26,762	99.7	26,220	99.7	0.267
Wound Dehiscence						
	Yes	28	0.1	25	0.1	
	No	26,802	99.9	26,275	99.9	0.734
Postoperative Pneumonia						
	Yes	990	3.7	1075	4.1	
	No	25,840	96.3	25,225	95.9	**0.018**
Reintubation						
	Yes	267	1	291	1.1	
	No	26,563	99	26,009	98.9	0.208
Postoperative Pulmonary Embolism						
	Yes	201	0.7	252	1	
	No	26,629	99.3	26,048	99	**0.009**
Postoperative Ventilator >48 h						
	Yes	131	0.5	124	0.5	
	No	26,699	99.5	26,176	99.5	0.78
Postoperative UTI						
	Yes	989	3.7	982	3.7	
	No	25,841	96.3	25,318	96.3	0.771
Postoperative CVA						
	Yes	203	0.8	187	0.7	
	No	26,627	99.2	26,113	99.3	0.538
Postoperative Cardiac Arrest						
	Yes	170	0.6	174	0.7	
	No	26,660	99.4	26,126	99.3	0.688
Postoperative Myocardial Infarction						
	Yes	509	1.9	605	2.3	
	No	26,321	98.1	25,695	97.7	**0.001**
Post-Operartive Bleeding Transfusion > 2						
	Yes	5212	19.4	5087	19.3	
	No	21,618	80.6	21,213	80.7	0.807
Postoperative DVT						
	Yes	234	0.9	279	1.1	
	No	26,596	99.1	26,021	98.9	**0.026**
Postoperative Sepsis						
	Yes	261	1	265	1	
	No	26,569	99	26,035	99	0.685
Postoperative Septic Shock						
	Yes	166	0.6	158	0.6	
	No	26,664	99.4	26,142	99.4	0.79
Return to Operating Room						
	Yes	627	2.3	615	2.3	
	No	26,203	97.7	25,685	97.7	0.991
Reoperation due to Hip Fracture						
	Yes	451	1.7	438	1.7	
	No	26,379	98.3	25,862	98.3	0.889
Unplanned Readmission						
	Yes	2018	7.5	2061	7.8	
	No	24,812	92.5	24,239	92.2	0.173
30-Day Mortality						
	Alive	25,487	95	24,811	94.3	
	Dead	1343	5	1489	5.7	**0.003**
Discharge Destination						
	Home	5644	21.4	7286	28.3	
	Facility	20,122	76.3	17,840	69.3	
	AMA	39	0.1	51	0.2	
	Expired	557	2.1	583	2.3	**<0.001**
		**Mean**	**Standard Deviation**	**Mean**	**Standard Deviation**	
Length of total hospital stay		4.67	14.8	4.57	15.9	0.463
Total operation time (minutes)		66.68	39.4	67.35	38.9	**0.047**
Days from hospital admission to operation		1.15	3.1	1.16	3.3	0.935

### 2.5. Statistical Analysis

The *χ*^2^ test of independence was used to compare the categorical variables. Two-sided independent sample *t*-test was used to compare the means of continuous variables. Logistic regression model was constructed to run a multivariable analysis with categorical pre-operative and postoperative variables achieving significance on univariable analysis.

The variable ‘Race’ was not included in our multivariable model. Inclusion of ‘Race’ in multivariable model resulted in unexpected singularities in the Hessian matrix.

## 3. Results

There were a total of 26,830 cases sampled for 2019 and 26,300 cases sampled for 2020 (Table 1).

### 3.1. Demographics (Table 1)

There was no significant change in age (*p* = 0.853) or gender (*p* = 0.356) distribution of patients undergoing proximal femoral fracture surgery before COVID-19 pandemic (2019) and during COVID-19 pandemic (2020). However, a statistically significant increase in racial distribution of Asian Pacific Islanders and a decrease in Caucasians was observed among patients with proximal femoral fractures in 2020 (*p* < 0.001).

### 3.2. Pre-Operative Factors (Table 1)

A statistically significant decrease was seen in the number of ‘independent’ patients from 2019 (78.5%) to 2020 (77.2%) (*p* < 0.001). Similarly, there was a decrease observed in the number of patients with chronic wounds from pre-COVID-19 (2019) to COVID-19 (2020) time period (*p* = 0.002). More than 10% loss of body weight was seen in only 2.4% of the patients during 2019 as compared to 4% of the patients during 2020 (*p* < 0.001). There were only 87.3% of patients with no signs of sepsis before the proximal femoral fracture surgery in 2019 as compared to 86.1% of patients in 2020 (*p* = 0.002). A statistically significantly higher number of cases were designated as emergent during 2020 (*p* < 0.001).

### 3.3. Intraoperative Factors

A higher ASA classification (*p* = 0.014) and a lower proportion of patients receiving general anesthesia (*p* < 0.001) was seen during COVID-19 pandemic (2020). A higher proportion of patients underwent surgical procedure ‘IM Nail placement’ during 2020 (45.2% vs. 43.9%). Additionally, surgical procedures ‘hemiarthroplasty’ (10.6% vs. 9.6%) and ‘ORIF intertrochanteric, peri-trochanteric, and sub-trochanteric with plates and screws’ (6.9% vs. 6%) were more commonly seen during 2019 (*p* < 0.001). There was an increased mean total operation time during 2020 (67.35) when compared to 2019 (66.68). This finding achieved border line statistical significance with *p* = 0.047 and is of limited clinical significance.

### 3.4. Postoperative Factors (Table 2)

A higher proportion of patients developed postoperative acute renal failure during the COVID-19 pandemic (0.9%) as compared to 2019 (0.7%). This finding achieved statistical significance with *p*-value = 0.006. A similar increase was seen in patients developing postoperative pneumonia (*p* = 0.018), DVT (*p* = 0.026), PE (*p* = 0.009) and acute myocardial infarction (MI) (*p* = 0.001). A higher proportion of patients was discharged home rather than another health facility during 2020 (28.3%) as compared to 2019 (21.4%) (*p* < 0.001). A higher 30-day post-op mortality of 5.7% was also observed during 2020 as compared to 5% during 2019 (*p* = 0.003).

### 3.5. Multivariable Analysis (Table 3)

A multivariable logistic regression model of statistically significant categorical variables on univariable analysis is presented in Table 3. Pre-operative factors: ‘Emergent’ status of the surgery (*p* < 0.001), ≥10% loss of body weight (*p* < 0.001), and postoperative factors: acute renal failure (0.012), PE (0.043), MI (0.005), DVT (0.044), discharge destination ‘Home’ (*p* < 0.001) and leaving ‘AMA’ (*p* = 0.027) and higher rate of 30-day mortality (*p* = 0.001) were statistically significantly associated with the year 2020 (pandemic year). The presence of an open wound (*p* < 0.001), ‘independent’ functional status (*p* < 0.001) and surgical procedures ‘hemiarthroplasty’ (*p* = 0.002) and ‘ORIF intertrochanteric, peri-trochanteric, and sub-trochanteric with plates and screws’ (*p* < 0.001) were statistically significant associated with pre-COVID-19.

**Table 3 jcm-11-06778-t003:** Multivariable analysis and logistic regression.

Multivariable Analysis					
Logistic Regression		n	OR	95% CI	*p*-Value
*Year of Diagnosis*	*Dependent Variable*				
	*2019*	*26,075*		*Reference*	
	*2020*	*25,513*			
≥10% loss of body weight					
	Yes	1630	1.669	1.506–1.850	**<0.001**
	No	49,958		Reference	
Presence of wound					
	Yes	1404	0.832	0.747–0.927	**<0.001**
	No	50,184		Reference	
Pre-Operative Sepsis					
	None	44,721	0.927	0.391–2.198	0.863
	SIRS	6614	1.027	0.433–2.436	0.953
	Sepsis	232	1.022	0.415–2.517	0.962
	Septic Shock	21		Reference	
Emergent					
	Yes	16,630	1.1	1.059–1.143	**<0.001**
	No	34,958		Reference	
Functional Status					
	Independent	40,356	0.883	0.802–0.973	**0.012**
	Partially Dependent	9440	0.954	0.861–1.056	0.362
	Fully Dependent	1792		Reference	
ASA					
	Class I	480	0.824	0.507–1.34	0.436
	Class II	8797	0.946	0.602–1.487	0.809
	Class III	31,983	1.076	0.686–1.688	0.751
	Class IV	10,251	1.091	0.695–1.714	0.704
	Class V	77		Reference	
Anesthesia					
	General	36,123	3.501	0.391–31.372	0.263
	Regional	10,764	3.948	0.44–35.389	0.22
	Other	4703		Reference	
Acute Renal Failure					
	Yes	413	1.292	1.061–1.1574	**0.011**
	No	51,175		Reference	
Postoperative Pulmonary Embolism					
	Yes	425	1.219	1.003–1.481	**0.047**
	No	51,163		Reference	
Postoperative Myocardial Infarction					
	Yes	1036	1.192	1.051–1.351	**0.006**
	No	50,552		Reference	
Postoperative DVT					
	Yes	495	1.192	1.051–1.351	**0.049**
	No	51,093		Reference	
30-Day Mortality					
	Alive	48,811	0.849	0.768–0.939	**0.001**
	Dead	2777		Reference	
Discharge Destination					
	Home	12,817	1.693	1.446–1.983	**<0.001**
	Facility	37,569	1.089	0.934–1.27	0.276
	AMA	90	1.658	1.061–2.59	**0.026**
	Expired	1112		Reference	
Surgical Procedure					
	Hemiarthroplasty	5161	0.843	0.757–0.939	**0.002**
	Total Hip Arthroplasty	3014	0.889	0.791-1	0.049
	ORIF Neck or Prosthetic	15,272	0.955	0.866–1.053	0.354
	ORIF IT, Peritroch, Subtroch with plates and screws	3257	0.782	0.697–0.878	**<0.001**
	IM Nail for IT, Peritroch, Subtroch	23,041	0.983	0.893–1.082	0.726
	Other	1843		Reference	

## 4. Discussion

This study utilized the ACS NSQIP PUF from year 2019 and 2020 to compare the demographics, preoperative, intraoperative, 30-day postoperative characteristics and 30-day post-op mortality for patients with proximal femoral fractures before and during the COVID-19 pandemic. To our knowledge, this is the first investigation of its kind, comparing a national sample of proximal femoral fracture patients before and during the pandemic. Our findings confirm higher 30-day post-op mortality during the pandemic year, a finding shared by others [6,7,10,18,19,21]. The current study highlights significant differences in patient characteristics and outcomes during the pandemic. The findings of this study are novel as this is the first study to report impact of COVID-19 on care of proximal femoral fractures sampled from the US national data. We believe that these findings will provide the fundamentals to institute policies and changes to equip and prepare the current health care system to face and tackle a similar challenge in future.

Using the logistic regression, we were able to ascertain independent preoperative and postoperative variables associated with the pandemic. Proximal femoral fracture surgery was more likely to be declared ‘Emergent’ status during the year 2020. Although this finding has not been previously reported in the literature, a case with an emergent status is more likely to be authorized, given the shortage and diversion of resources during COVID-19. A weight loss of ≥10% of body weight was also found to be independently statistically significant for the pandemic year, another finding unique to our analysis. Patients with proximal femoral fractures during pandemic were less likely to be functionally independent and less likely to have an open wound. Boukebous et al. presented an analysis of proximal femoral fracture patients treated at a level 1 trauma center in France during the pandemic [10]. They did not find any difference in IADL (Instrument of activities of daily living) between 2019 and 2020. However, IADL was an independent predictor of all-cause 30-day mortality in their analysis [10].

Patients undergoing surgery for proximal femoral fractures were more likely to develop acute renal failure during pandemic. Levitt et al. presented an analysis of N3C (National COVID-19 cohort collaborative) and found a statistically significant increase in acute kidney injury in COVID-19 positive patients [19]. Similar findings were also reported by Hall et al. [15]. In the current study, patients with proximal femoral fractures had a higher likelihood of DVT, PE and MI during the pandemic. A higher incidence of thromboembolic events has also been reported by others [11,15,19]. Galivanche et al. reported a higher incidence of myocardial infarction among COVID-19 positive patients undergoing surgical fixation [11]. Our investigation has also shown that patients were more likely to receive ‘hemiarthroplasty’ or ‘ORIF intertrochanteric, peri-trochanteric, and sub-trochanteric with plates and screws’ in 2019 on a multivariable analysis. This finding has not been previously reported in the literature. Our dataset lacked the details about surgical approach for hemiarthroplasty, however, a recent study emanating from Italy reported improved oxygenation in lateral decubitus position as compared to supine position [27].

Our investigation has shown an increase in proportion of Asian Pacific islanders and a decrease in proportion of Caucasians with proximal femoral fractures during 2020. Egol et al. also reported an observed increase in proportion of Asian Pacific islander patients with hip fractures as compared to the corresponding months of the previous year [6]. However, due to a smaller sample size, their finding was not statistically significant. A statistically significant increase in Asian Pacific Islander patients has not been previously reported in the literature. However, caution should be exercised in interpreting this finding. As mentioned earlier, NSQIP collects and reports a sample of cases from over 680 hospitals in the U.S. [23]. The sampling methods for data collection have been standardized [23]. In the past, others have utilized the NSQIP data to validate the demographic distribution in prospective clinical trials [28]. However, considering that NSQIP collects a sample of actual data, this finding needs additional validation. Such validation would be ideally obtained by using a national population-based registry to calculate the incidence of hip fracture during 2020 for different racial groups.

### Limitation

The major limitation of our analysis is the lack of data regarding covid infection. ACS NSQIP^®^ has yet to release the data regarding COVID-19 status. Thus, a direct comparison of covid positive patients with COVID-19 negative patients was not possible. We have only presented the data during the COVID-19 pandemic and compared it to the data before the pandemic. Another limitation of our study is a lack of survival analysis to determine the prognostic factors. COVID-19 positivity has been widely implicated as a significant prognostic factor in the literature [6,15,19,29]. An analysis of prognostic factors in the absence of COVID-19 status would be inherently biased. Thus, an analysis of prognostication has not been included.

## 5. Conclusions

In this retrospective NSQIP database study comparing hip fracture surgery patients prior to and during the COVID-19 pandemic, we identified an increased proportion of Asian Pacific islanders patients with hip fractures during the pandemic. Additionally, this study confirmed previously identified increased postoperative mortality among hip fracture patients during the pandemic [6,7,10,18,19,21]. Our study has also demonstrated that despite the stress of COVID-19 on the healthcare system, time from hospital admission to OR for proximal femur fractures did not increase. Surgical procedures of ‘hemiarthroplasty’ and ‘ORIF intertrochanteric, peri-trochanteric and sub-trochanteric with plates and screws’ were performed less often during the pandemic. The study sample is derived from more than 680 hospitals across the US and thus is representative of the national data.

## Data Availability

The source data is publicly available at ACS website: https://www.facs.org/quality-programs/data-and-registries/acs-nsqip/participant-use-data-file/ (accessed on 7 October 2022).

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
