# Peer review of "Traumatic Proximal Femoral Fractures during COVID-19 Pandemic in the US: An ACS NSQIP® Analysis"

_jcm, 2022, doi:10.3390/jcm11226778_

Round 1

Reviewer 1 Report

1. The abstract section should be enhanced to include quantitative data.

2. As the conclusion of your abstract, please provide a "take-home" message.

3. Rearrange keywords alphabetically.

4. It is encouraged not used abbreviations in the keywords section.

5. Novelty in the current study's is too weak. The past has seen an extensive study of a lot of written material. It is required to provide more details for more explanation about the present novel in the introductory section.

6. Line 35, dot (.) should be after reference.

7. In order to highlight the gaps in the literature that the most recent research aims to fill, it is crucial to review the benefits, novelty, and limitations of earlier studies in the introduction.

8. The explanation is weak, more extending explanation is mandatory. Since it is discussed related hip fracture, hip joint replacement surgery and/or total hip prosthesis need to be mentioned. It is a crucial issue that authors should provide in the introduction and/or discussion section. The suggested reverence should be adopted as follows: Ammarullah, M. I.; Santoso, G.; Sugiharto, S.; Supriyono, T.; Kurdi, O.; Tauviqirrahman, M.; Winarni, T. I.; Jamari, J. Tresca Stress Study of CoCrMo-on-CoCrMo Bearings Based on Body Mass Index Using 2D Computational Model. Jurnal Tribologi 2022, 33, 31–8. https://jurnaltribologi.mytribos.org/v33/JT-33-31-38.pdf

9. To help the reader grasp the study's workflow more easily, the authors could include more visuals to the materials and methods section in the form of figures rather than sticking with the text that now predominates.

10. An evaluation of the findings with similar past research is essential.

11. The conclusion section needs to explain further research.

12. The reference is recommended to be enriched with literature from five years ago. MDPI reference is strongly recommended.

13. The authors were encouraged to proofread their work due to grammatical problems and linguistic style.

14. A graphical abstract is suggested to be included in the submission after peer review.

Author Response

Reviewer 1:

Thank you so much for your review and useful suggestions. Mentioned below are our responses.

  1. The abstract section should be enhanced to include quantitative data.

Thank you so much for your useful suggestion. There is a word limit of 200 words for the abstract and the abstract needs to be structured in the sections of background, methods, results and discussion. We have done our best to include what we feel as the most important quantitative data in terms of p-values. Unfortunately, we can not include any more quantitative date given the constraints of word limit.

  1. As the conclusion of your abstract, please provide a "take-home" message.

The last sentence of the abstract is the ‘take home’ message outlining the differences in the outcomes and preoperative characteristics.

  1. Rearrange keywords alphabetically.

Thank you; the keywords have been rearranged.

  1. It is encouraged not used abbreviations in the keywords section.

Thank you, the abbreviations used in the keywords section are COVID-19 and ACS NSQIP. We feel that these abbreviations are more familiar to audience than the actual spelled out phrases: Coronavirus disease 2019 and American College of Surgeons National Surgical Quality Improvement Program. The phrases are defined in the abstract section. We would like to keep it the abbreviations.

  1. Novelty in the current study's is too weak. The past has seen an extensive study of a lot of written material. It is required to provide more details for more explanation about the present novel in the introductory section.

This is the only study, to the best of our knowledge, looking at the representative national data across US reporting the preoperative characteristics of patient with proximal femur fractures during COVID-19 pandemic comparing it to the year before and reporting statistically significant differences. There is no national data study available from the US on proximal femoral fracture during the pandemic. COVID-19 pandemic had unprecedented stress on the US health system. This is the novelty of our study.

  1. Line 35, dot (.) should be after reference.

Thank you so much, the change has been made.

  1. In order to highlight the gaps in the literature that the most recent research aims to fill, it is crucial to review the benefits, novelty, and limitations of earlier studies in the introduction.

Thank you. We agree with the reviewer. The relevant previous studies have been cited and discussed in the introduction section from lines 42-52, third paragrapgh.

  1. The explanation is weak, more extending explanation is mandatory. Since it is discussed related hip fracture, hip joint replacement surgery and/or total hip prosthesis need to be mentioned. It is a crucial issue that authors should provide in the introduction and/or discussion section. The suggested reverence should be adopted as follows: Ammarullah, M. I.; Santoso, G.; Sugiharto, S.; Supriyono, T.; Kurdi, O.; Tauviqirrahman, M.; Winarni, T. I.; Jamari, J. Tresca Stress Study of CoCrMo-on-CoCrMo Bearings Based on Body Mass Index Using 2D Computational Model. Jurnal Tribologi 2022, 33, 31–8. https://jurnaltribologi.mytribos.org/v33/JT-33-31-38.pdf

Thank you again for your suggestion. We respectfully disagree with the suggestion by the reviewer. Our study is reporting the preoperative characteristics and outcomes of patients with proximal femoral fractures from ACS NSQIP database during the COVID-19 pandemic. Information regarding the bearing surfaces is lacking in the database and is not relevant to the objective of our study.

  1. To help the reader grasp the study's workflow more easily, the authors could include more visuals to the materials and methods section in the form of figures rather than sticking with the text that now predominates.

Thank you for your suggestions. We have included extensive tables as part of the study. It is a very simple study design of retrospective review of the ACS NSQIP database.

  1. An evaluation of the findings with similar past research is essential.

We have compared and contrasted our findings in the discussion section in detail with the previous similar studies across the world.

  1. The conclusion section needs to explain further research.

Thank you for your comment. The conclusion section highlights the key important findings of our research paper. We are not sure what exactly do the reviewer mean by ‘explain further research’.

  1. The reference is recommended to be enriched with literature from five years ago. MDPI reference is strongly recommended.

Thank you for your comment. Our paper is focused proximal femoral fracture during COVID-19 pandemic. The only papers that have cited and discussed relevant research were published after the pandemic. We have added reference # 25 from MDPI lines 214-216.

  1. The authors were encouraged to proofread their work due to grammatical problems and linguistic style.

Thank you, the manuscript has been re-read for grammatical errors and changes have been tracked.

  1. A graphical abstract is suggested to be included in the submission after peer review.

Thank you for your suggestions. However, we currently don’t have a graphical abstract.

Reviewer 2 Report

This article aims to describe the impact of COVID-19 disease on the treatment and outcomes in patients with proximal femoral fractures.

I read the article with interest, the title is well thought out and faithfully reflects the content of the study.

The topic of the study is interesting and evaluates parameters in orthopedic fractures.

There is a difference in the covid disease description throughout the text: COVID, covid, COVID-19.

I recommend writing it COVID-19 or COVID in capital letters.

According to my opinion, some minor changes are needed to be considered suitable for publication:

-The abstract is adequately developed and is useful to frame the characteristics and purpose of the study.

The introduction is complete and comprehensive.

Materials and methods are well structured. If you use an acronym the first time, please express it (ORIF).

The results reflect the purpose of the study.

The discussion is sufficiently developed and conclusion is supported by results.

- Introduction:

Well written, concise. Clearly describes the study objectives and background.

I suggest results were statistically significant could be briefly introduced at least by macroareas.

- Method:

The paper describes the differences between pre and post covid period. The use of a large database improves the results of the paper. It would be preferable to begin the methods with a description of the type of work, inclusion/exclusion criteria, characteristics of participants, whether or not ethics committee approval or registration with international database of studies such as "clinicaltrials.gov" is needed. You could rearrange the paragraph "ACS-NSQIP®" or introduce the methods and then the dataset.

If you use an acronym the first time, please express it (COPD, CHF, HTN, CPT codes).

Better define activities of daily living (ADLs), describing the scores of the different categories to avoid bias.

-Results:

There were a total of 26,380 cases sampled for 2019 and 26,300 cases sampled for 2020 116 (Table 1)”.

The sample is different from abstract: “A total of 26,830 and 26,300 patients 12 sustaining PFF and undergoing surgical treatment were sampled during 2019 and 2020, respectively”. Please correct.

Table 1 is missing years between the two groups as is Table 2?

- Discussion: This section is too short. It is worthwhile to recall main findings in this study before stating conclusion.

This is one of the first papers with a large sample from US databases. Better explain the role these findings might play in the literature. Why this work deserves to be published, what it adds to researchers, and its future implications.

A significant study published on JCM-MDPI showing the difference in surgical approach for hemiarthroplasty on proximal femur fractures in patients with COVID-19, highlighted the role of decubitus on respiratory parameters (DOI: 10.3390/jcm11164785).

- Conclusions are consistent with the evidence and arguments presented.

“bOur study” line 230, please correct.

Please review the whole text, avoiding grammatical and linguistic errors.

Author Response

Reviewer 2:

This article aims to describe the impact of COVID-19 disease on the treatment and outcomes in patients with proximal femoral fractures.

I read the article with interest, the title is well thought out and faithfully reflects the content of the study.

The topic of the study is interesting and evaluates parameters in orthopedic fractures.

There is a difference in the covid disease description throughout the text: COVID, covid, COVID-19.

I recommend writing it COVID-19 or COVID in capital letters.

 Thank you so much for your review and useful suggestions. We have changed to COVID-19 as per your recommendations.

According to my opinion, some minor changes are needed to be considered suitable for publication:

 Thank you.

-The abstract is adequately developed and is useful to frame the characteristics and purpose of the study.

The introduction is complete and comprehensive.

Materials and methods are well structured. If you use an acronym the first time, please express it (ORIF).

The results reflect the purpose of the study.

The discussion is sufficiently developed and conclusion is supported by results.

- Introduction:

Well written, concise. Clearly describes the study objectives and background.

I suggest results were statistically significant could be briefly introduced at least by macroareas.

Thank you for your suggestions. We have highlighted the p-values in the abstract section. The results are described in the results section and summarized in the conclusion section. We’ll be grateful if reviewer can kindly elaborate what do they mean by ‘briefly introduced at least by macroareas’. 

- Method:

The paper describes the differences between pre and post covid period. The use of a large database improves the results of the paper. It would be preferable to begin the methods with a description of the type of work, inclusion/exclusion criteria, characteristics of participants, whether or not ethics committee approval or registration with international database of studies such as "clinicaltrials.gov" is needed. You could rearrange the paragraph "ACS-NSQIP®" or introduce the methods and then the dataset.

Thank you so much. We have added the lines 60-63 in the Methods section as suggested by the reviewer.

If you use an acronym the first time, please express it (COPD, CHF, HTN, CPT codes).

Thank you, we have defined the acronyms.

Better define activities of daily living (ADLs), describing the scores of the different categories to avoid bias.

Thank you for the comment. This is a category (IADL) defined by ACS NSQIP and detail of how the scores are assigned has been described by ACS in detail. We have used the numbers reported by NSQIP. It is beyond the scope of current paper to describe the NSQIP scheme for scoring in IADL.

 -Results:

There were a total of 26,380 cases sampled for 2019 and 26,300 cases sampled for 2020 116 (Table 1)”.

The sample is different from abstract: “A total of 26,830 and 26,300 patients 12 sustaining PFF and undergoing surgical treatment were sampled during 2019 and 2020, respectively”. Please correct.

Thank you so much. We have corrected it in the first sentence of the results section.

Table 1 is missing years between the two groups as is Table 2?

 Thank you so much. We have included the years in the Table 1.

- Discussion: This section is too short. It is worthwhile to recall main findings in this study before stating conclusion.

This is one of the first papers with a large sample from US databases. Better explain the role these findings might play in the literature. Why this work deserves to be published, what it adds to researchers, and its future implications.

A significant study published on JCM-MDPI showing the difference in surgical approach for hemiarthroplasty on proximal femur fractures in patients with COVID-19, highlighted the role of decubitus on respiratory parameters (DOI: 10.3390/jcm11164785).

 Thank you so much. We have included the following in the first paragraph from 184-190 lines in the discussion section.

‘The current study highlights significant differences in patient characteristics and out-comes during the pandemic. The findings of this study are novel as this is the first study to report impact of COVID-19 on care of proximal femoral fractures sampled from the US national data. We believe that these findings will provide the fundamentals to institute policies and changes to equip and prepare the current health care system to face and tackle a similar challenge in future.’

We have added the reference mentioned by the reviewer to our manuscript. Reference 25, lines 214-216.

- Conclusions are consistent with the evidence and arguments presented.

Thank you so much.

“bOur study” line 230, please correct.

Thank you so much. We have corrected that.

 Please review the whole text, avoiding grammatical and linguistic errors.

We have reviewed the manuscript for grammatical errors.

Round 2

Reviewer 1 Report

Reviewers greatly appreciate the efforts that have been made by the author to improve the quality of their articles after peer review. I reread the author's manuscript and further reviewed the changes made along with the responses from previous reviewers' comments. Unfortunately, the authors failed to make some of the substantial improvements they should have made making this article not of decent quality with biased, not cutting-edge updates on the research topic outlined. In addition, the author also failed to address the previous reviewer's comments, especially on comments number 5 (not use “to the best of our knowledge, but proof it with literature searching from main database such as Scopus, Web of Science, and PubMed), 8 (extending explanation and suggested literature not incorporated properly), and 14 (the authors should provide it as suggested).  Thank you very much for the opportunity to read the author's current work.

Author Response

Response to the reviewer 1:

Thank you again for your review and comments.

The reviewer has mentioned that our manuscript requires extensive improvement in English language and style. Can the reviewer please provide further details regarding this comment as careful review of our manuscript does not indicate the need for extensive editing of the English language.

The reviewer has indicated that all aspects of the manuscript ‘must be improved’, without detailing specific areas to improve. May the reviewer please clarify the specifics of this request so we can appropriately address the comments in our paper.

Regarding the term ‘to the best our knowledge’, we performed an extensive literature search using the databases listed. We are unaware of additional articles detailing the effect of COVID-19 on the surgical treatment of patients with proximal femur fractures in the US. If we are missing an important paper or dataset, we would be happy to include this in our manuscript. We have added two further references from Europe Ref # 17 & 20.

We have respectfully indicated that we do not agree with the reviewer’s suggestion # 8. The present study reviews preoperative characteristics and outcomes of patients with proximal femur fractures from the ACS NSQIP database during the COVID-19 pandemic. A discussion of Tresca stress and its effect on Metal on metal hip implants is certainly outside the scope of this study and not relevant to our discussion.

We are unable to provide a graphical abstract. This is a descriptive study; as such, we do not have  figures to include in a graphical abstract.

We are thankful for each reviewer spending the time to thoroughly evaluate our manuscript. We ask for a review of reviewer 1’s comments and suggestions as we find the suggested revisions unfitting to the manuscript. Thank you for your consideration and time.

Reviewer 2 Report

The manuscript was revised according to the suggestions.

Author Response

Thank you for your time and useful suggestions.